# Image Muting of Mixed Precipitation to Improve Identification of Regions of Heavy Snow in Radar Data

Laura M. Tomkins[1], Sandra E. Yuter[1,2], Matthew A. Miller[2], and Luke R. Allen[1]

[1]Center for Geospatial Analytics, North Carolina State University, Raleigh, NC, 27695, USA
[2]Department of Marine, Earth and Atmospheric Science, North Carolina State University, Raleigh, NC, 27695, USA

**Correspondence:** Laura Tomkins (lmtomkin@ncsu.edu)

**Abstract.**

In winter storms, enhanced radar reflectivity is often associated with heavy snow. However, some higher reflectivities are the result of mixed precipitation including melting snow. The correlation coefficient (a dual-polarization radar variable) can identify regions of and mixed precipitation, but this information is usually presented separately from reflectivity. Especially under time pressure, radar data users can mistake regions of mixed precipitation for heavy snow because of the high cognitive load associated with comparing data in two fields while simultaneously attempting to discount a portion of the high reflectivity values. We developed an image muting method for regional radar maps that visually deemphasizes the high reflectivity values associated with mixed precipitation. These image muted depictions of winter storm precipitation structures are useful for analyzing regions of heavy snow and monitoring real-time weather conditions.

## 1 Introduction

Weather radar data from ground-based scanning radars are crucial for monitoring the location, intensity, and evolution of storms. Winter storms in mid-latitude regions often contain subregions with rain, mixed precipitation, and snow that move and evolve over the storm lifetime (Schultz et al., 2019). Higher radar reflectivity values are generally associated with heavier precipitation. But the transition among rain, partially melted snow, and snow precipitation types creates a challenge when interpreting radar reflectivity because volumes with melting precipitation have higher reflectivities than volumes with the equivalent precipitation mass of only ice hydrometeors or only liquid hydrometeors (Vivekanandan et al., 1994; Straka et al., 2000; Rauber and Nesbitt, 2018).

In particular, the changes in phase from ice to partially melted ice and then to rain modify the dielectric constant of the particles so that volumes with the same precipitation mass per unit volume can have different reflectivity values (Battan, 1973). When analyzing banded snow features in winter storms, areas of mixed precipitation can be distracting and misleading (e.g. Picca et al., 2014). We define mixed precipitation as precipitation that includes combinations of rain or freezing rain, snow, sleet, and partially-melted snow.

Regions of mixtures of precipitation types can be identified with the dual-polarization radar variable known as the correlation coefficient ($\rho_{HV}$) (Table 1; e.g. Vivekanandan et al., 1994; Straka et al., 2000; Kumjian, 2013a). Correlation coefficient is a

statistical measure of how consistent the shapes and sizes of particles are within a radar resolution volume (Rauber and Nesbitt, 2018). This variable is insensitive to radar calibration and yields comparable values for the same set of hydrometeors across radar networks with identical hardware and signal processing methods. Correlation coefficient is approximately one in regions with single hydrometeor types (e.g. only rain or only snow) and decreases in regions where there is an increasing diversity of hydrometeor orientations and shapes (e.g. mixed precipitation such as rain with snow and/or partially melted ice) (Giangrande et al., 2008; Rauber and Nesbitt, 2018). Additionally, correlation coefficient can have low values in various types of ground clutter and is used in identifying non-meteorological echo (e.g. Zrnić et al., 2006; Alku et al., 2015; Kumjian, 2013b).

While correlation coefficient is insensitive to radar power calibration, it does suffer from other data quality problems. Artificially lower $\rho_{HV}$ values can occur along radials downrange of sharp gradients in differential phase (Ryzhkov, 2007). With increasing range from a radar, radar resolution volume size increases and signal to noise ratio (SNR) decreases. For example near the melting layer, larger radar resolution volumes are more likely to have non-uniform beam filling than smaller radar resolution volumes. In theory, non-uniform beam filling of a radar resolution volume would tend to decrease correlation coefficient (Ryzhkov, 2007). However, the current method used to compute correlation coefficient in U.S. NEXRAD operational radars yields increased values with decreasing SNR (Ivić, 2019). Unlike radars that transmit at horizontal and vertical polarizations, the NEXRAD radar transmits at a single polarization oriented at 45 degrees which reduces the overall sensitivity of the radar and in conditions with canted, oriented ice can reduce the correlation coefficient (Rauber and Nesbitt, 2018). In practice, the impact of SNR tends to be much more prevalent than non-uniform beam filling. This suggests that the SNR effect masks most of the effects of non-uniform beam filling in NEXRAD correlation coefficient data quality. Dual polarization radar variable data quality problems are more pronounced when there are mismatched antenna patterns in the horizontal and vertical polarizations (Bringi and Chandrasekar, 2001) which are more common in operational radars than research radars.

Since reflectivity, correlation coefficient, and hydrometeor types are usually presented as separate products (NOAA, 2017), someone wanting to discern regions of heavy snow versus mixed precipitation in a winter storm needs to toggle back and forth among different products or overlay them. Neural science studies show that switching between sources of information increases the cognitive load of a task (Sweller et al., 2011; Harrower, 2007). Keeping track of changing shapes of moving objects is particularly challenging (Suchow and Alvarez, 2011). Integrating related material and removing irrelevant material is essential for maximizing understanding and learning (Mayer and Moreno, 2003; Sweller et al., 2011; Harrower, 2007).

In order to reduce the cognitive load associated with analyzing precipitation structures in reflectivity, we propose a new visualization technique we refer to as "image muting". Image muting aids interpretation of sequences of radar data in movie loops. We plot the reflectivities using a perceptually uniform, color-blind-friendly color scale and the subset of reflectivity values corresponding to mixed precipitation using a gray scale of matching perceptual lightness. This visualization does not remove areas of melting but rather "mutes" them, making the regions stand out less than the snow-only or rain-only portions of the storm. Work by Calvo et al. (2021) demonstrates how making small changes in climate visualizations can reduce the cognitive load and support analysis and potential decision making.

Our image muting technique is described in detail in Sect. 2, and applications of our technique are presented in Sect. 4.

## 2 Methods

To demonstrate the methodology, we used Level-II data from several National Weather Service (NWS) Next-Generation Radar (NEXRAD) network radars in the northeast United States (US) that were obtained from the NOAA Archive on Amazon Web Services (Ansari et al., 2018). Complete volume scans are available from each radar approximately every 5 to 10 minutes. This technique can be applied to any radar data set that has both reflectivity and correlation coefficient fields.

### 2.1 Regional Mapping

We combine data from several radars to create regional radar maps utilizing functions in the open source Python Atmospheric Radiation Measurement (ARM) Radar Toolkit developed by the Department of Energy ARM Climate Research Facility (Py-ART; Helmus and Collis, 2016). We first extract the first 0.5° elevation angle plan-position indicator (PPI) from each volume scan. We do not interpolate to a constant altitude in order to preserve as much fine-scale detail in the reflectivity and correlation coefficient structures as possible. We include only data within 200 km range from a radar as this is sufficient for combining data from multiple radars in much of the continental US without substantial gaps and constrains the beam center to be below 4 km altitude above radar level. The polar coordinate data from each individual radar are interpolated using Cressman weighting (Cressman, 1959) to a Cartesian grid covering our geographic region of interest. Before interpolating, we convert the reflectivity from units of dBZ to units of $\mathrm{mm}^6\mathrm{m}^{-3}$ because interpolating in linear reflectivity units provides a more accurate representation of the polar data (Warren and Protat, 2019). We interpolate each polar radar object used in the regional map to the same Cartesian grid with 2 km grid spacing. For the northeast US regional maps shown in this paper, the regional grid is 1201 km x 1201 km. We convert the reflectivity back to dBZ after the interpolation step. Finally, to combine data from all the radars into a single object, we designate a "central radar" to stitch all the other radars to. For storms in the northeast US, we use the Long Island, NY (KOKX) radar as the central radar. For each volume scan at KOKX, we find the closest time from the other radars (within 8 minutes). For grid points where coverage from adjacent radars overlaps, we use data from the radar with the maximum reflectivity value and its corresponding correlation coefficient value. Use of the maximum reflectivity value means adjacent points can be from 0.5° elevation angles from different radars yielding discontinuities in altitude of up to 4 km. Since our main research application is identifying snow bands and lighter versus heavier regions of snow, having adjacent points not continuous in altitude was an acceptable trade off. Before plotting the fields, we despeckle the data to remove areas of echo that are less than 20 $\mathrm{km}^2$.

### 2.2 Identification of mixed precipitation

In effect, we are implementing a hydrometeor identification for only mixed precipitation. We simplify the radar data visualization by choosing this one hydrometeor category to *deemphasize* in the reflectivity field. We identify grid points where the hydrometeors are partially melted and/or mixed rain and snow, where the $\rho_{HV}$ is below a threshold of 0.97, and where the reflectivity values is greater than or equal to 20 dBZ. We used 0.97 following Giangrande et al. (2008) who found that the correlation coefficient for dry snow exceeded this value. Adding the criterion of reflectivity $\geq$ 20 dBZ was essential in distin-

guishing regions of melting or mixed precipitation that could be confused with heavy snow from regions of light precipitation with noisy, unreliable $\rho_{HV}$ values. The 0.97 $\rho_{HV}$ and 20 dBZ thresholds are consistent with Griffin et al. (2020) who used $\rho_{HV}$ to detect melting layers in radar data. We note that not all clutter points are removed in our regional maps which can have low values of $\rho_{HV}$ and may show up as stationary features in animations of image muted maps.

Any method relying on a particular variable as input will not work well when there are data quality problems with that variable. Data quality problems with correlation coefficient along radials downrange of sharp gradients in differential phase will yield sporadic image muted areas radial to the radar that will not move consistently with the advection of locally enhanced reflectivity bands within the storm. Regions of speckled image muting based on the method described here could either be a result of small spatial scale variations in the melting of snow or noise in the correlation coefficient field related to low signal to 100   noise ratios which are more common at farther ranges from the radar (Ivić, 2019).

The inputs and outputs for image muting from a coastal winter precipitation event on 07 February 2020 are shown in Fig. 1. Information from regional maps of the radar reflectivity field (Fig. 1a) and the correlation coefficient field (Fig. 1b) are combined. We show an intermediate stage (Fig. 1c) illustrating the pragmatic importance of the using both the correlation coefficient and reflectivity criteria. $\rho_{HV}$ values $\leq 0.97$ often occur toward the edges of the individual radar echo domains 105   where the beam is $> \sim 3$ km altitude and in winter storms very likely to be only snow (green region in Fig. 1c). We infer that the reflectivity $< 20$ dBZ is too low to reliably indicate mixed precipitation that can be mistaken for heavy snow. The areas in gray represent regions where the $\rho_{HV} \leq 0.97$ and the reflectivity is $\geq 20$ dBZ, where melting is likely to be present and where we mute the reflectivity. Dark blue colors in Fig. 1c are where the correlation coefficient is $> 0.97$, indicative of uniform precipitation types. The final image muted reflectivity product (Fig. 1d) uses a gray scale to deemphasize the subset 110   of reflectivity values where it is likely to be mixed precipitation. This example shows two linear features in central New York that could be misinterpreted as purely snowbands when analyzing the reflectivity alone (white ovals in 1a). The animation of this figure (Video Supplement Animation-Figure-1) for the time period 12:00:00 to 15:00:00 UTC shows how the mixed precipitation region covers portions of the high reflectivity bands in Fig. 1a as the bands move eastward. The image muted reflectivity helps users focus on regions of the storm that are not affected by mixed precipitation. We experimented with trying 115   to distinguish the rain-only from the snow-only regions but found that there was insufficient information in the dual-polarization radar variables to do this reliably without data on air temperature. Air mass and frontal boundaries can cause freezing level heights to vary sharply within winter storms unlike warm-season precipitation.

## 3   Evaluation with independent data

Vertical cross-sections from airborne radar data provide an opportunity to evaluate the identification of melting regions in 120   ground-based scanning radar data in fine detail. Figure 2 shows an image muted regional map corresponding to a science flight during the NASA Investigation of Microphysics and Precipitation for Atlantic Coast-Threatening Snowstorms (IMPACTS) 2020 field project (McMurdie et al., 2022). Reflectivity from the nadir-pointing ER-2 X-band Doppler Radar (EXRAD; Heymsfield et al., 1996) along the flight track (green line) in Fig. 2a is shown in Fig. 2b. The gray region in the image muted regional

map indicates a quasi-linear region of mixed precipitation extending through eastern New York up to Vermont and New Hampshire (Fig. 2a) between areas of primarily snow (to the northwest in upstate New York) and primarily rain (to the southeast over southern New England). Eastward of 175 km along the flight transect in Fig. 2b, there is a clear melting layer signature in the NASA EXRAD data starting near the surface and rising to about 2 km above surface level (ASL) (represented by the enhanced region of higher reflectivity). The melting layer can also be observed with other variables from the same transect presented in Fig. 3. In particular, the linear depolarization ratio from the ER-2 cloud radar shows the structure of the melting layer very well (Fig. 3d). Under the melting layer, the values of downward pointing Doppler velocity < -4 ms$^{-1}$ indicate the rain layer. The position of the transition between snow and rain in the vertical cross-section is consistent with the edge of the gray area in Fig. 2a. An animated version of this figure shows the timing as the ER-2 aircraft transects through the image muted portion of the regional map (Video Supplement Animation-Figure-2). As the airplane reaches around 175 km in the transect, one can see that the height of the NEXRAD radar beam used to create the regional map (black X in Fig. 2b) begins to intersect the melting layer.

Information to further evaluate the timing and location of the melting and mixed precipitation is available from time series of precipitation from surface sensors. Figure 4 shows hourly time series of precipitation types at several NWS Automated Surface Observing Systems (ASOS) weather stations (letters in Fig. 2a). The surface observations and timing of precipitation transitions align well with the evolution and movement of the storm (Fig. 2 and 3). For the hour of 16:00:00 UTC, Syracuse Hancock International Airport (KSYR) is reporting snow, Albany International Airport (KALB) is reporting rain, Greater Binghamton, NY (KBGM) is reporting snow, and Westchester County Airport (KHPN) is reporting rain. The ASOS time series for KBGM also indicates the hour when rain transitioned to mixed (11:00:00 UTC) and mixed transitioned to snow (15:00:00 UTC) (Fig. 4c). These surface data are consistent with the locations of the muted precipitation (Video Supplement Animation-Figure-2).

## 4   Application to RHIs

Information on the 3D geometry of melting regions can be obtained by applying the image muting technique to range-height indicator (RHI) scans constructed from a full volume scan from ground-based scanning radars. These examples illustrate the often complex layering within coastal winter storms where portions of the warmer air masses (> 0° C) slide over colder air masses (< 0° C). Figure 5 is from the KOKX radar during a winter storm on 08 February 2013. The green line in the PPIs corresponds to the azimuth used to create the RHIs (Fig. 5a,b). Rather than a simple flat or tilted melting layer, this storm had a 3D "arc-like" mixed precipitation structure (Fig. 5c,d). The temperature field along the RHI from the ERA5 reanalysis data shows the associated vertical temperature structure and the 0° C isotherm (Fig. 5e; Hersbach et al., 2020). Below 2 km ASL, the temperature is mostly above freezing, which corresponds well to the top of the melting in the RHI panels (Fig. 5c,d,e). There appears to be an intrusion of colder air around 0.5 km ASL (0–30 km horizontal) that is likely contributing to the arc-like feature seen in the RHI panels (Fig. 5c,d,e). Animations of panels a through d of Fig. 5 show the complex horizontal pattern as the features evolve and move (Video Supplement Animation-Figure-5). The structure of the melting layer in this example is also discussed in Griffin et al. (2014).

An example from the Philadelphia, PA (KDIX) radar during a winter storm on 01 December 2019 is presented in Fig. 6. This storm exhibited an interesting "collapsing" signature in the correlation coefficient and image muted reflectivity PPI fields in northern New Jersey (Fig. 6a,b). The RHI panels intersect the feature and show a sharp drop in melting layer altitude around the 80 km range from the radar (Fig. 6c,d). The temperature field from the ERA5 reanalysis shows an elongated layer of above freezing temperatures around 2 km ASL and another area of above freezing temperatures below 1 km ASL between 0 and 50 km away from the radar (Fig. 6e). It is likely that the ERA5 data are too coarse to fully represent the complex temperature structure as suggested by the radar RHIs. Animations of panels a through d of Fig. 6 show the initiation of this feature and how it evolves (Video Supplement Animation-Figure-6).

Users should use caution interpreting features at longer ranges from the radar where $\rho_{HV}$ suffers from quality issues related to low signal to noise ratio. For example, in Figure 6, the speckled muting beyond 100 km range of the radar is likely the result of the superposition of an increase in correlation coefficient associated with low signal to noise ratio and a decrease associated with melting. The animation of this figure (Video Supplement Animation-Figure-6) illustrates that the concentric speckled region remains approximately stationary to the radar and hence can be visually distinguished from advecting reflectivity bands.

## 5  Summary

The proliferation of weather radar web interfaces and mobile apps has made operational radar data easily accessible to a wide range of users with varying levels of radar data interpretation expertise. People who are well versed in the subtle nuances of interpreting weather radar data represent only a subset of research meteorologists and an even smaller subset of the broader set of radar data users which includes emergency managers, TV weathercasters, and airport operators.

Users of weather radar data associate areas of higher reflectivities with heavier precipitation. In winter storms, linear features of localized enhanced reflectivity are associated with heavy snow bands and contribute to snow accumulation forecast uncertainties (e.g. Novak et al., 2008; Ganetis et al., 2018). But regions of mixed precipitation can exhibit higher reflectivities often without the higher precipitation rates or equivalent liquid water content. For winter storm analysis, it is important to distinguish between locally enhanced reflectivity associated with increases in ice mass and reflectivity from melting. Fortunately, mixed precipitation often has a low correlation coefficient ($< 0.97$) which in combination with reflectivities $\geq 20$ dBZ can be used to distinguish higher reflectivity regions that are and are not heavy snow (Giangrande et al., 2008).

Typically, radar reflectivity and hydrometeor identification are presented as separate products (Rauber and Nesbitt, 2018; Bringi and Chandrasekar, 2001; NOAA, 2017). When these products are separate, a user examining an evolving winter storm needs to simultaneously examine synced sequences of maps and mentally keep track of the moving positions of higher reflectivity features relative to the hydrometeor type signatures.

We developed image muting, which reduces the visual prominence of the reflectivities within the mixed precipitation features in winter storms that can be mis-identified as heavy snow. Reflectivities corresponding to the mixed precipitation features are deemphasized using a gray scale and the regions with just snow and just rain are depicted in a corresponding full-color scale. We tuned the thresholds used for identification of mixed precipitation areas using a combination of detailed vertical cross-

sections from research aircraft radar, reconstructed RHIs from ground-based scanning radars, and surface weather stations observed precipitation types. Users could apply this visualization technique using operational hydrometeor classification as an input and mute other specific regions depending on the application.

Enhanced reflectivity bands that are snow or contain mixed precipitation will generally move consistently with the advection of other reflectivity features rather than being fixed either concentrically or radially to the radar position. Hence, our image muting method is best used as part of movie loop sequences rather than as individual images. Image muted movie loops will help reduce the error associated with misinterpreting radar reflectivity products during winter storms. Users examining an image muted 2D map movie loop can more easily distinguish the locations of heavy snow and mixed precipitation as compared to having to consult separate movie loops. Monitoring where transitions from rain to mixed precipitation and mixed precipitation to snow are present and where they are likely to move can aid in assessing expected impacts of winter weather.

The method to detect melting regions is not perfect in large part since such algorithms are limited by the input data quality. For U.S. NEXRAD data, without improvements in the data quality of $\rho_{HV}$, detection of melting regions particularly at farther ranges will be more speckled than at closer ranges. If the signal to noise ratio field is made available it can be used to filter out questionable $\rho_{HV}$ values and improve the detection of melting regions. Users are advised to utilize movie loops to assess the time and spatial continuity when distinguishing band-like enhanced reflectivity features corresponding to heavy snow bands from those that include melting.

The image muting visualization technique can be applied to a wide variety of applications. Any data display that suffers from potential misinterpretation could benefit from image muting portions of the data to de-emphasize subregions in the plot.

*Code and data availability.* Data: The NWS NEXRAD Level-II data used in Figs. 1, 2, 5, and 6 can be accessed from the National Centers for Environmental Information (NCEI) at https://www.ncei.noaa.gov/products/radar/next-generation-weather-radar. The NASA IMPACTS radar data used in Fig. 2 can be accessed at https://ghrc.nsstc.nasa.gov/uso/ds_details/collections/impactsC.html. The NWS ASOS surface station data used to create Fig. 4 can be accessed from NCEI at https://www.ncei.noaa.gov/products/land-based-station/automated-surface-weather-observing-systems. The ERA5 reanalysis data used in Figs. 5 and 6 can be accessed from the Copernicus Climate Change Service (C3S) Climate Data Store at https://www.ecmwf.int/en/forecasts/datasets/reanalysis-datasets/era5.

Code: We submitted functions to make image muted maps to the Py-ART GitHub repository (Helmus and Collis, 2016) to facilitate use of this technique by others. They were accepted and released in Py-ART version 1.11.8. The Py-ART function used to create the figures in the paper can be accessed via https://arm-doe.github.io/pyart/API/generated/pyart.util.image_mute_radar.html. An example of how to use the function is provided here: https://arm-doe.github.io/pyart/examples/plotting/plot_nexrad_image_muted_reflectivity.html#sphx-glr-examples-plotting-plot-nexrad-image-muted-reflectivity-py.

*Video supplement.* List of animations with captions and filenames

All animations can be viewed at: https://av.tib.eu/series/1228. Individual animations can be viewed by following the DOI URL.

Animation-Figure-1: Animated plot of image muting processing components for a radar regional map from 12:00:00 to 15:00:00 UTC on 07 February 2020. (a) Radar reflectivity (dBZ) field. (b) Correlation coefficient field. (c) Categories indicating regions that meet the following conditions: correlation coefficient > 0.97 (dark blue), correlation coefficient ≤ 0.97 and reflectivity < 20 dBZ (green), and correlation coefficient ≤ 0.97 and reflectivity ≥ 20 dBZ (gray). (d) Final image muted product combining color scale for reflectivities in snow and rain regions with gray scale to mute reflectivities in mixed precipitation regions. (goes with Fig. 1). Title: 07 February 2020 image muting example Filename: fig01_animation.mp4 DOI: https://doi.org/10.5446/57311

Animation-Figure-2: Animated plot of image muted regional map with detailed vertical cross-section from NASA ER-2 X-band Doppler radar during a NASA IMPACTS science mission on 07 February 2020. At 16:09:10 UTC, the aircraft is located at the transition between snow and melting precipitation in the radar regional map. (a) Image muted reflectivity valid at 16:11:03 UTC with the ER-2 flight leg (green line), aircraft location corresponding to time shown in bottom panel is at the arrow head along the leg. Locations of ASOS observations in Fig. 4 are annotated with stars and black labels. (b) Vertical cross-section of reflectivity from NASA EXRAD radar with current aircraft location near the top of the vertical green line. Time at right corresponds to aircraft position. The black X indicates the height of the point in panel a that varies along the 0.5° elevation angle scans used to construct the regional maps. (goes with Fig. 2). Title: 07 February 2020 NASA IMPACTS transect comparison Filename: fig02_animation.mp4 DOI: https://doi.org/10.5446/57312

Animation-Figure-5: Animated plot of image muted regional map with reconstructed RHIs and reanalysis temperature vertical cross-section from KOKX radar on 08 February 2013. (a) Correlation coefficient and (b) image muted reflectivity (dBZ) 0.5° elevation angle PPI plots for KOKX radar valid valid 21:00:00 UTC 08 February to 00:00:00 UTC 09 February 2013. Green line in (a) and (b) indicates location of reconstructed RHI cross-sections from (c) correlation coefficient and (d) image muted reflectivity. (e) ERA5 reanalysis temperature cross-section interpolated to the plane of the RHI. Black line in panel e indicates 0° C isotherm. (goes with Fig. 5). Title: 08 February 2013 KOKX RHI comparison Filename: fig05_animation.mp4 DOI: https://doi.org/10.5446/57313

Animation-Figure-6: Animated plot of image muted regional map with reconstructed RHIs and reanalysis temperature vertical cross-section from KDIX radar on 01 December 2019. (a) Correlation coefficient and (b) image muted reflectivity (dBZ) 0.5° elevation angle PPI plots for KDIX radar valid 15:00:00 to 20:00:00 UTC on 01 December 2019. Green line in (a) and (b) indicates location of reconstructed RHI cross-sections from (c) correlation coefficient and (d) image muted reflectivity. (e) ERA5 reanalysis temperature cross-section interpolated to the plane of the RHI. Black line in panel e indicates 0° C isotherm. (goes with Fig. 6). Title: 01 December 2019 KDIX RHI comparison Filename: fig06_animation.mp4 DOI: https://doi.org/10.5446/57314

*Author contributions.* LMT and SEY conceptualized the project and designed the methodology with input from MAM. LMT wrote the software and created the visualizations with input from SEY, MAM, and LRA. LMT prepared the manuscript with all authors contributing to review and editing.

*Competing interests.* The authors declare that they have no conflict of interest.

*Acknowledgements.* The authors express their appreciation to V. Chandrasekar (Colorado State University) and Scott Ellis (NCAR) for their insights into radar data quality issues. Special thanks also to McKenzie Peters, Anya Apontes-Torres, and Jordan Fritz for their assistance in data processing, to Kevin Burris and Rachel Kennedy for providing feedback on the manuscript, and to Christina Cartwright for editing the manuscript. This work was supported by the National Science Foundation (AGS-1347491 and AGS-1905736) and the National Aeronautics 260 and Space Administration (80NSSC19K0354) and the Center for Geospatial Analytics at North Carolina State University.

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

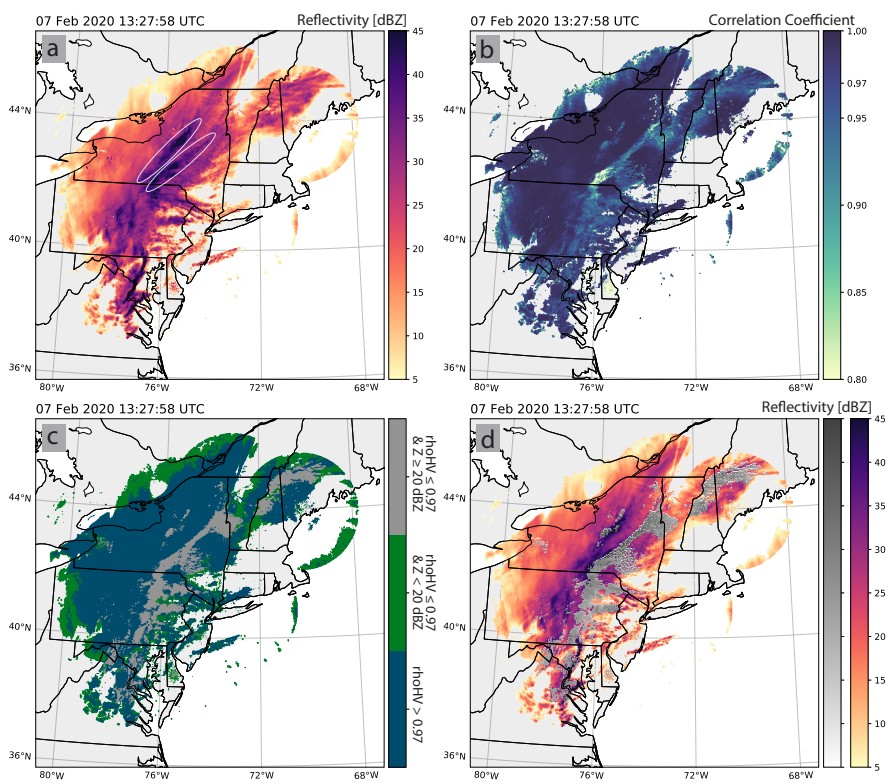

**Figure 1.** Image muting processing components for a radar regional map from 07 February 2020 at 13:27:58 UTC. (a) Radar reflectivity (dBZ) field. (b) Correlation coefficient field. (c) Categories indicating regions that meet the following conditions: correlation coefficient > 0.97 (dark blue), correlation coefficient ≤ 0.97 and reflectivity < 20 dBZ (green), and correlation coefficient ≤ 0.97 and reflectivity ≥ 20 dBZ (gray). (d) Final image muted product combining color scale for reflectivities in snow and rain regions with gray scale to mute reflectivities in mixed precipitation regions. White ovals in (a) indicate banded features discussed in text. An animated version of this figure is in Video Supplement Animation-Figure-1.

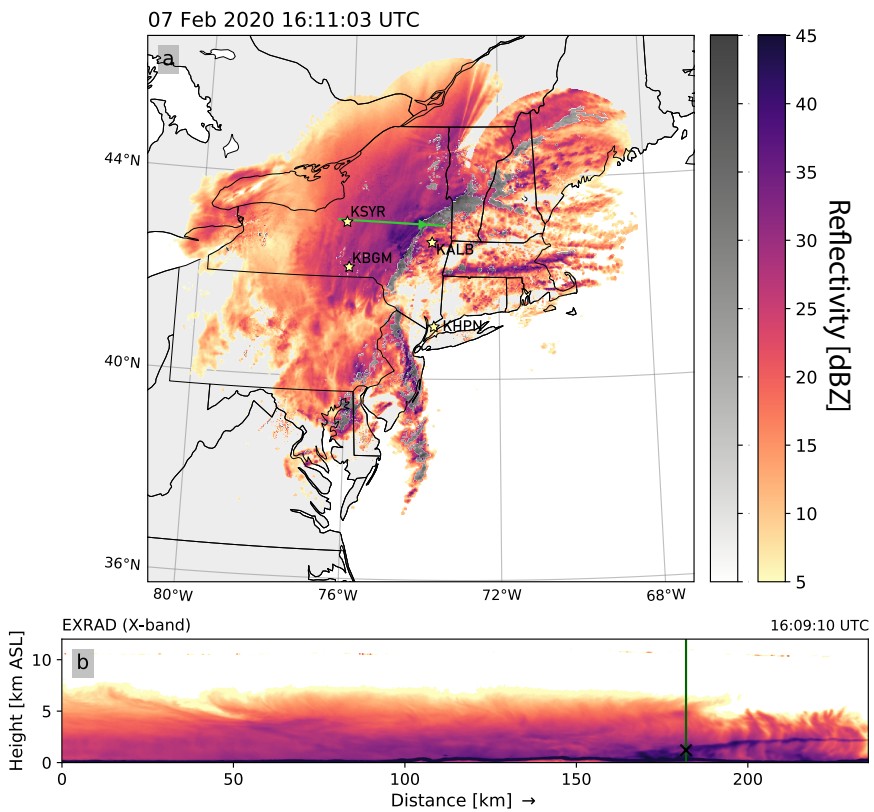

**Figure 2.** Comparison of image muted regional map with detailed vertical cross-section from NASA ER-2 X-band Doppler radar during a NASA IMPACTS science mission on 07 February 2020. At 16:09:10 UTC, the aircraft is located at the transition between snow and melting precipitation in the radar regional map. (a) Image muted reflectivity valid at 16:11:03 UTC with the ER-2 flight leg (green line), aircraft location corresponding to time shown in bottom panel is at the arrow head along the leg. Locations of ASOS observations in Fig. 4 are annotated with stars and black labels. (b) Vertical cross-section of reflectivity from NASA EXRAD radar with current aircraft location near the top of the vertical green line. Time at right corresponds to aircraft position. The black X indicates the height of the point in panel a that varies along the 0.5° elevation angle scans used to construct the regional maps. An animated version of this figure is in Video Supplement Animation-Figure-2.

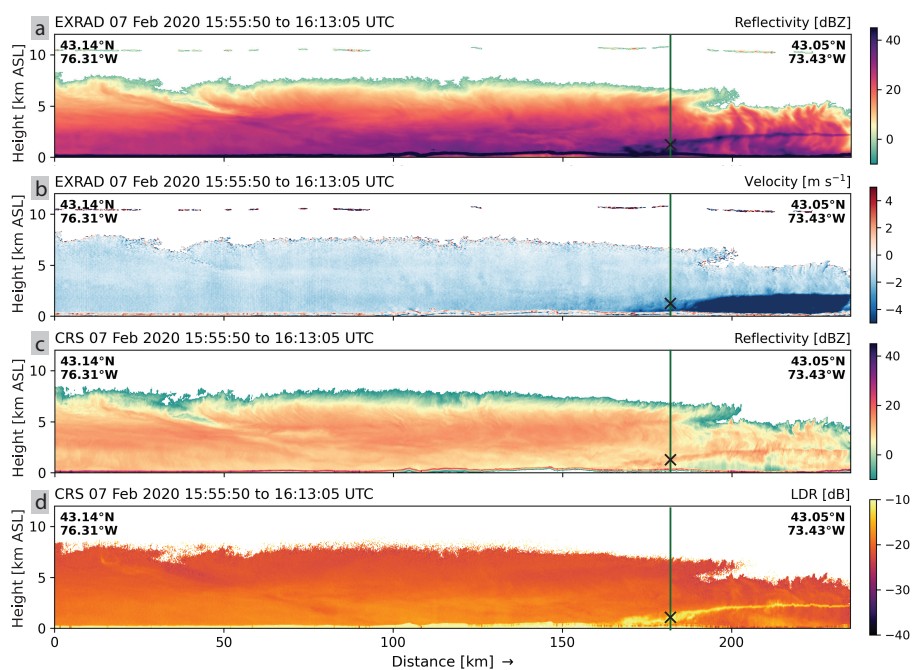

**Figure 3.** Vertical cross-sections of (a) reflectivity and (b) vertical velocity from NASA ER-2 EXRAD radar and (c) reflectivity and (d) linear depolarization ratio (LDR) from NASA ER-2 CRS radar coincident with vertical cross section in 2. Green line indicates current aircraft location and black X indicates the height of the point in 2a that varies along the 0.5° elevation angle scans used to construct the regional maps.

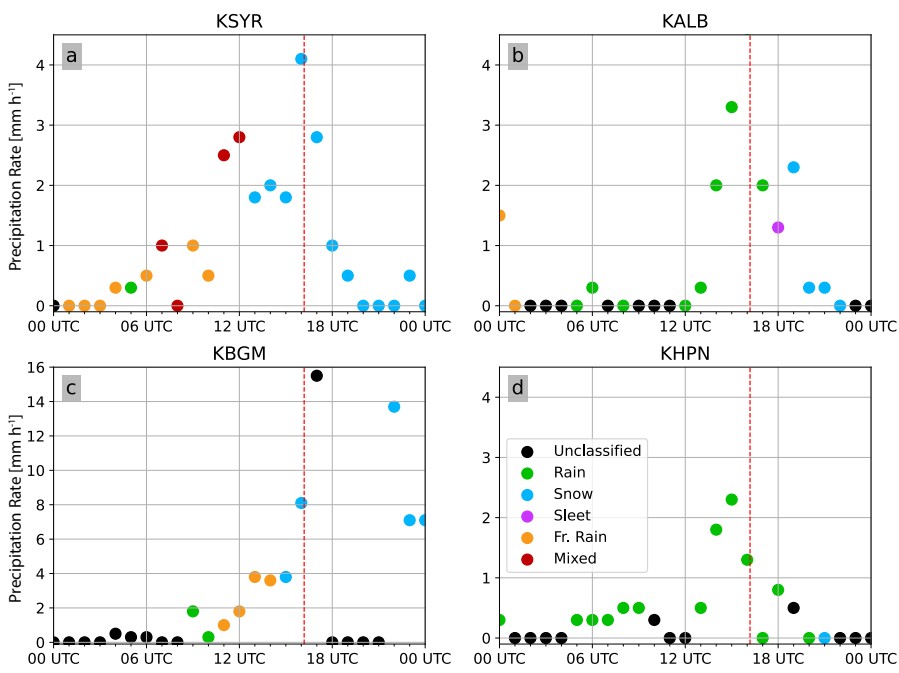

**Figure 4.** Hourly ASOS precipitation rate and type [mm h$^{-1}$] reports for 07 February 2020 from (a) KSYR, (b) KALB, (c) KBGM and (d) KHPN. Colors indicate precipitation type as in legend in panel d. Red dashed line indicates 16:09:11 UTC, highlighted in Fig. 2. The y-axis range is larger in panel c compared to other panels.

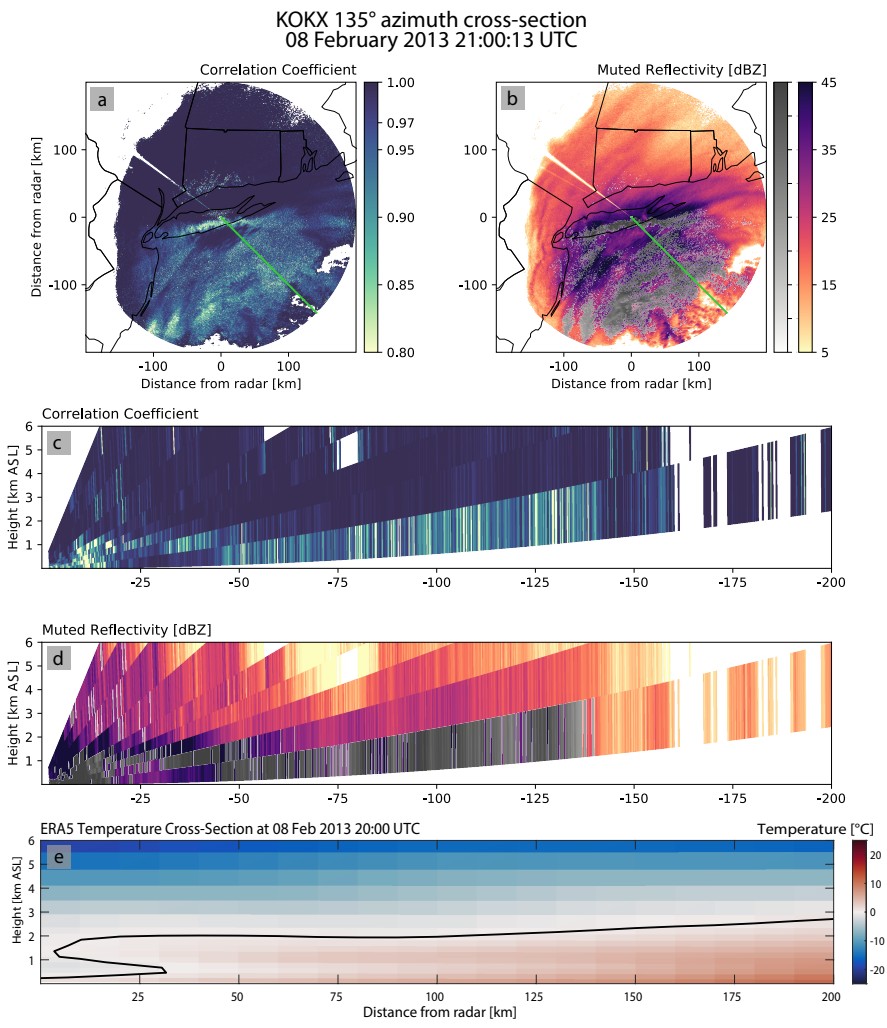

**Figure 5.** Comparison of image muted regional map with reconstructed RHIs and reanalysis temperature vertical cross-section from KOKX radar on 08 February 2013. (a) Correlation coefficient and (b) image muted reflectivity (dBZ) 0.5° elevation angle PPI plots for KOKX radar valid 08 February 2013 21:00:13 UTC. Green line in (a) and (b) indicates location of reconstructed RHI cross-sections from (c) correlation coefficient and (d) image muted reflectivity. (e) ERA5 reanalysis temperature cross-section interpolated to the plane of the RHI. Black line in panel e indicates 0° C isotherm. An animated version of this figure is in Video Supplement Animation-Figure-5.

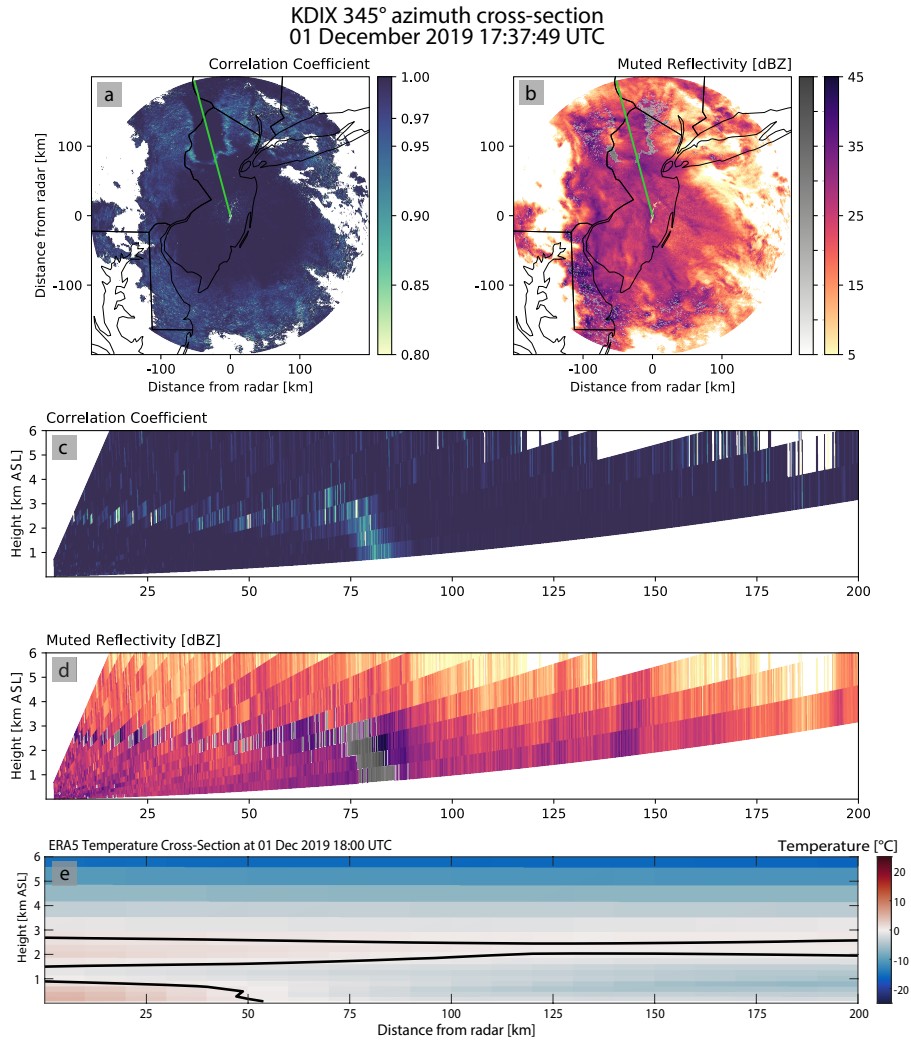

**Figure 6.** Comparison of image muted regional map with reconstructed RHIs and reanalysis temperature vertical cross-section from KDIX radar on 01 December 2019. (a) Correlation coefficient and (b) image muted reflectivity (dBZ) 0.5° elevation angle PPI plots for KDIX radar valid 01 December 2019 17:37:49 UTC. Green line in (a) and (b) indicates location of reconstructed RHI cross-sections from (c) correlation coefficient and (d) image muted reflectivity. (e) ERA5 reanalysis temperature cross-section interpolated to the plane of the RHI. Black line in panel e indicates 0° C isotherm. An animated version of this figure is in Video Supplement Animation-Figure-6.

| Description | Increase number of ice particles in snow | Increase size of ice particles in snow | Mixtures of partially melted ice, ice, and rain |
| --- | --- | --- | --- |
| Change to water substance mass per unit volume | Increases | Increases | No change |
| $\rho_{HV}$ value | $\sim 1$ | $\sim 1$ | $< 0.97$ |

**Table 1.** Correlation coefficient values associated with physical mechanisms that increase radar reflectivities when Z > 20 dBZ and other conditions are held constant.