# Peer review of "Image Muting of Mixed Precipitation to Improve Identification of Regions of Heavy Snow in Radar Data"

_Atmospheric Measurement Techniques, 2022_

## Author Response (AR1)

**Author response**

Quoted text in revised manuscript

**Tomkins et al. (2022) review response**

RC1: 'Comment on amt-2022-160', Anonymous Referee #1, 11 Jun 2022

This paper describes a visual technique to remove regions of melting hydrometeors in radar displays in winter storms. The technique allows forecasters and researchers to readily identify regions of heavy snowfall and associated hazards. This paper describes one implementation of such a product using NEXRAD data in the USA and corroborates the technique with some independent data. Overall the paper does a good job describing the technique and its general application. The contribution is worthy of publication, however I wish to suggest some improvements to the paper to help the reader through the reasoning behind it as well as comment on the particular application of the technique. Thus, I am returning the paper for major revisions.

L24: It might be worth noting that correlation coefficient is insensitve to radar power calibration issues, and intrinsic measurements of correlation coefficient should be consistent amongst radars with similar hardware and signal processing techniques (such as NEXRAD). Some intrinsic differences/biases could exist with radars with different transmission and signal processing techniques (number of pulses; antenna crosspolar performance; radome effects; determination of noise and receiver accuracy; spectral vs pulse pair processing).

**We agree and have added the following text to the Introduction:**

"This variable is insensitive to radar calibration and yields comparable values for the same set of hydrometeors across radar networks with identical hardware and signal processing methods." *(lines 26-27)*

L53:

(a) Polarimetric data quality is known to degrade with range, causing correlation coefficient values to decrease uniformly due to factors such as non-uniform beam filling. Does this impact the classification of pixels as "mixed" uniformly with range?

**We investigated further by seeking out expert opinions from radar experts Dr. Chandraskar at CSU and Scott Ellis at NCAR and obtained more information that resulted in the following text to Introduction.**

"With increasing range from a radar, radar resolution volume size increases and signal to noise ratio (SNR) decreases. For example near the melting layer, larger radar resolution volumes are more likely to have non-uniform beam filling than smaller radar resolution volumes. In theory, non-uniform beam filling would tend to decrease correlation coefficient (Ryzhkov, 2007). Unlike radars that transmit at horizontal and vertical polarizations, the NEXRAD radar transmits at a single polarization oriented at 45 degrees. The current method used to compute correlation coefficient in US NEXRAD operational radars yields increased values with decreasing SNR (Ivić, 2019). In practice, the impact of SNR tends to be much more prevalent than nonuniform beam filling. This suggests that the SNR effect masks most of the effects of non-uniform beam filling in NEXRAD correlation coefficient data quality." *(lines 32-29)*

**We are not defining a new method to detect all melting regions. Rather, we seek to de-emphasize melting regions that could be misinterpreted as heavy snow. Hence, applying our muting technique to regions where the correlation coefficient is < 0.97 and reflectivity is greater than 20 dBZ was appropriate. We clarified our reasoning for the Z threshold in the text.**

"Adding the criterion of reflectivity  20 dBZ was essential in distinguishing regions of melting or mixed precipitation that could be confused with heavy snow from regions of light  precipitation with noisy, unreliable HV values." *(lines 85-87)*

(b) In addition to mixed populations of hydrometeors, correlation coefficient also is lowered in regions of partial beam blockage and mainlobe and sidelobe clutter (terrain being one factor leading to this). Do you notice any stationary regions where mixed precipitation is more likely to be classified due to these effects?

**In some of the older data (late 1990s and early 2000s) we do see some stationary features in the regional maps likely associated with the clutter types you mention. In the paper, we rely on the US NWS algorithms for the removal of ground clutter. Before plotting, we despeckle the fields (areas of echo less than 20 km$^2$ are removed) which removes some of the remaining clutter points. In a movie presentation the clutter points that remain are stationary and have a higher reflectivity than the background so they are easy to visually discount. To remove these residual points it would be necessary to implement our own ground clutter removal algorithm but this is a separate topic outside of the scope of this paper.**

**We have added a sentence with citations to the Introduction to describe more fully that low correlation coefficient can also occur in ground clutter.**

"Additionally, correlation coefficient can have low values in various types of ground clutter and is used in identifying non-meteorological echo (e.g. Zrnić et al., 2006; Alku et al., 2015; Kumjian, 2013b)." *(lines 30-31)*

**We have also added a sentence describing the despeckling step:**

"Before plotting the fields, we despeckle the data to remove areas of echo that are less than 20 km$^2$." *(lines 78-79)*

L71: What is the sensitivity of choosing a value of 0.97? In a fuzzy logic scheme, which is the current state of the art method for hydrometeor classification, uniform thresholds are not used, rather many variables are used and the "winning" hydrometeor classification is then selected. Can you comment on why a more sophisticated scheme was not used? Or even the operational hydrometeor classification in the NEXRAD? Perhaps it could be stated that the technique could be applied to any effort to censor data that might confound the user (clutter, biological scatter, partial beam blockage, non-uniform beam filling, etc.)

**We did sensitivity testing on the threshold and found that 0.97 worked best overall and was supported by Giangrande et al. (2008).  Fuzzy logic hydrometer algorithms cluster points associated with multiple variables (Bringi and Chandrasekar 2001).**

**We found that a simple correlation coefficient threshold worked well for our intended purpose which is to de-emphasize melting that could be misinterpreted as heavy snow.**

**The current US NWS hydrometeor identification algorithm often struggles in winter storms. For example, Figure R1 shows a sequence of images comparing the hydrometeor classification and image muted reflectivity. While the NWS hydrometeor classification algorithm correctly identifies some of the areas with low RHOHV as wet snow, it also miscategorizes portions of the melting areas as graupel, big drops, and heavy rain. Additionally, there are "jumps" in spatial continuity of regions with a given classification. On balance, we determined that the NWS hydrometeor classification algorithm's performance in winter weather was not well suited to our purpose.**

**Image muting as a visualization method has many uses. A colleague who was inspired by our paper is using image muting for some their climate model visualizations. Ground clutter echoes are usually just removed from the radar display (e.g. https://www.canada.ca/en/environment-climate-change/services/weather-general-tools-resources/radar-overview/about-radar.html#toc2)**

Figure R1: Sequence of images from KBGM radar on 7 Feb 2020 valid 13:38 UTC (left), 13:44 UTC (center), and 13:50 UTC (right). Top panel is Hydrometeor

classification, bottom panel is image muted reflectivity

[Figure]

**We have added text to describe how this product could be used as an input to the technique:**

"Users could apply this visualization technique using operational hydrometeor classification as an input and mute other specific regions depending on the application." *(line 175-176)*

General comment: The video files in the supplement seem to suffer from video compression issues. If the authors could change their compression settings, that would be helpful to the reader.

**Thank you for the comment. The animations have been changed to their original quality in the portal.**

L81: "reflectivity < 20 dBZ is too low to reliably indicate mixed precipitation". Can you expound/give a physical basis for this? Is this due to the long ranges used in the analysis - a quick perusal of NEXRAD data shows very high values of correlation coefficient in reflectivities as low as 5 dBZ in drizzle?

**We have changed the sentence to the following:**

*"We infer that the reflectivity < 20 dBZ is too low to reliably indicate mixed precipitation that can be mistaken for heavy snow."* (line 95)

L104: Are there other parameters in the EXRAD data or other data collected aboard the aircraft to more reliably denote the melting layer? Was doppler velocity or linear depolarization ratio measured by any of the suite of radars on board?

**We have added an extra figure to the manuscript which shows several different fields from the same transect to better illustrate the melting layer and the rain layer beneath it (see Figure R2) with the following text.**

*"The melting layer can also be observed with other variables from the same transect presented in Fig. 3. In particular, the linear depolarization ratio from the ER-2 cloud radar shows the structure of the melting layer very well (Fig. 3d). Under the melting layer, the values of downward pointing Doppler velocity > -4 ms$^{-1}$ indicate the rain layer."* (lines 117-119)

Figure R2: Other fields from the ER-2 radars to depict the melting layer. Has been added to the manuscript as Figure 3.

[Figure]

RC2: 'Comment on amt-2022-160', Anonymous Referee #2, 22 Jun 2022

Overall Recommendation:

The authors present a new reflectivity visualization technique that aims to decrease cognitive load on radar analysts by muting reflectivity in areas classified as mixed precipitation. I think there is a possible application of this technique for those who would potentially struggle with appropriately diagnosing higher reflectivity associated with melting. However, operational meteorologists are generally trained in awareness of the brightband. Moreover, the spatiotemporal evolution of winter storms (relative to severe convective hazards) allows for more time to analyze multiple products, etc. Thus, I struggle to see a significant need in the operational community for such a tool, at least given some of the concerns/shortcomings I mention below (muting of important features, simple thresholds, etc.). I think this paper could be a worthy contribution to the literature if these concerns are appropriately addressed via either a) making a stronger argument for how this would be beneficial for operational meteorologists and/or b) framing it more for non-meteorologists. Additionally, shortcomings in the algorithm logic/performance need to be addressed.

General/Major Comments:

- Winter weather scenarios, while challenging in different ways from severe convective scenarios, tend to evolve on longer timescales than severe convective scenarios. Is the time pressure in winter mixed precipitation events that high to necessitate this image muting technique for operational forecasters? Is there evidence to support this claim that even experienced meteorologists are mistaking bright banding for heavier precipitation? Not saying it isn't occurring, but is it occurring enough to necessitate a new product? I would argue this is more of a training issue for forecasters vs the need for a separate product. That said, I could see more value in such a product being presented as a visualization tool for non-meteorologists in weather-sensitive fields (e.g., emergency management) or for the broader public, perhaps in weathercasts or in apps. I wonder if it might be beneficial to emphasize this technique as a presentation tool for non-meteorologists.

**Nowhere in the original manuscript did we state that this tool was needed by operational meteorologists. Only a subset of meteorologists work for national weather services.**

**The authors have personally witnessed many experienced meteorologists mistake bright band regions for heavy snow dozens of times over the years. Multiple incidents occurred during field projects as real-time radar data are being viewed to make time-sensitive aircraft deployment decisions. Additionally, we have seen numerous presentations at conferences and workshops over the years that make this mistake. In academia, this misinterpretation is more likely in less experienced analysts such as graduate students. In part, this common mistake motivated this work.**

**To clarify, we added material to describe our intended users in the Summary and in the Abstract replaced "even experienced meteorologists" with "radar data users".**

"The proliferation of weather radar web interfaces and mobile apps has made operational radar data easily accessible to a wide range of users with varying levels of radar data interpretation expertise. People who are well versed in the subtle nuances of interpreting weather radar data represent only a subset of research meteorologists and an even smaller subset of the broader set of radar data users which includes emergency managers, TV weathercasters, and airport operators." *(lines 155-158)*

"Especially under time pressure, radar data users can mistake regions of mixed precipitation for heavy snow because of the high cognitive load associated with comparing data in two fields while simultaneously attempting to discount a portion of the high reflectivity values." *(lines 4-7)*

- Section 2.2: I have concerns about the thresholding process. First, by filtering out light echoes (less than 20 dBZ), you do miss areas of mixed precipitation. I've witnessed scenarios with light crystals generated at low levels (e.g., top of the boundary layer) that then melt near the surface. In those instances, correlation coefficient (CC) can remain reliable, while still suggesting the presence of mixing. In these cases, a user of this product could think that these light echoes might be pure rain or snow, when it's mixed precip in reality. Moreover, what's the advantage of using flat CC/Z thresholds to identify mixed precip when other algorithms (like the 88D Hydrometeor Classification Algorithm) take into account more data in a more nuanced fashion? Would these not perform better at identifying mixed precipitation? Could you use such an algorithm as input into a more advanced muting technique?

**We are not defining a new method to detect all melting regions. Rather, we seek to de-emphasize melting regions that could be misinterpreted as heavy snow.**

**We found that a simple correlation coefficient threshold < 0.97 with the 20 dBZ Z threshold worked well for our intended purpose which is to de-emphasize melting that could be misinterpreted as heavy snow.**

**The current US NWS hydrometeor identification algorithm often struggles in winter storms. For example, Figure R1 shows a sequence of images comparing the hydrometeor classification and image muted reflectivity. While the NWS hydrometeor classification algorithm correctly identifies some of the areas with low RHOHV as wet snow, it also miscategorizes portions of the melting areas as graupel, big drops, and heavy rain. Additionally, there are "jumps" in spatial continuity of regions with a given classification. On balance, we determined that the NWS hydrometeor classification algorithm's performance in winter weather was not well suited to our purpose.**

**The hydrometeor classification could be used as an input to the muting technique. We have added the following text to the summary:**

*"Users could apply this visualization technique using operational hydrometeor classification as an input and mute other specific regions depending on the application." (line 175-176)*

Figure R1: Sequence of images from KBGM radar on 7 Feb 2020 valid 13:38 UTC (left), 13:44 UTC (center), and 13:50 UTC (right). Top panel is Hydrometeor classification, bottom panel is image muted reflectivity

[Figure]

- L78-81: I'm glad the algorithm isn't muting reflectivity at farther ranges, but I think the reasoning here is incorrect. Frequently, the reason lower CC is dominating at

these ranges (at least for the 88D network) is due to the radar sampling echoes at higher altitudes / colder temperatures, within the crystal generation region. The mixture of crystal habits is what's often driving CC downward. It isn't an unreliability of the signal. At a minimum, I think this is somewhat "getting it right for the wrong reasons." Most of the time, this crystal growth region should be characterized by relatively low Z, such that I don't think this would be a huge issue for the current design, but I think this needs to be corrected / clarified.

**Upon consultation with radar experts Dr. Chandrasekar (Colorado State University) and Dr. Scott Ellis (NCAR), we were made aware of a peculiar problem with the NWS NEXRAD algorithm to estimate correlation coefficient that is documented in Ivic (2019).**

**NEXRAD radars use a single transmission chain that emits energy at 45 deg polarization, and two separate receivers at H and V polarization. This hardware configuration yields dual polarization data that is inherently more noisy than research radars that transmit at H and V and receive at H and V polarization.**

**The current method used to compute correlation coefficient in US NEXRAD operational radars yields increased values with decreasing SNR. We have added mention of this issue and the citation to Ivic in the Introduction.**

**As an illustration of the problem with NEXRAD's RHOHV, please see example below, courtesy of Scott Ellis, which shows correlation coefficient values near 1 at farther ranges and low Z values where SNR is lower.**

[Figure]

Unlike many research radars, the NEXRAD products do not include a SNR field but a minimum detectable signal (MDS) can be estimated using the following formula:

*Zmin_detectable=dBZo + 20\*log10(rangeinkm)*

Where dBZo includes the hardware's contribution (radar system noise and receiver calibration) estimated at -44 dBm and a factor to account for non-hardware originated noise such as terrain, ground clutter, and clouds that can decrease SNR at low elevation angles.

Based on
https://www.roc.noaa.gov/WSR88D/PublicDocs/NOAA_Radar_Functional_Requirements_Final_Sept%202015.pdf

**5.4.6. Minimum detectable signal (MDS) and Sensitivity**

A low MDS is necessary to provide the sensitivity to detect weak, fine scale targets such as gust fronts and weak circulations, and to obtain valid velocity measurements from targets such as insects and moisture discontinuities. Radar sensitivity is closely related to the MDS. Higher sensitivity corresponds to an ability to detect weak signal strength features to longer ranges. The Threshold RFR is cited for a range of 50 km to provide reference point values of the required MDS.

**Threshold: Performance of WSR-88D**
- 0.0 dB Signal to Noise Ratio (SNR) for a -9.5 dBZe target at 50 km in short pulse and a -18.5 dBZe target at 50 km in long pulse

We estimate the NEXRAD hardware radar constant as -44 dBm and plot values of MDS for the non-hardware noise factor from 0 to 10 dBm in the first plot and the estimated SNR for a 20 dBZ echo as a function of range in the 2nd plot. (SNR=dBZ-MDS).

Liu et al (1994, JTECH, volume 11 pages 950-963) showed that to keep rhohv with +/- 0.01 you need an SNR of > about 20 dB. So the expectation is that the NEXRAD dual polarization variable values for a 20 dB echo will start to drop off in quality starting at about ~50, ~90 km or ~150 km range depending on how much non-hardware noise one wants to include in the estimate of MDS.

[Figure]

[Figure]

**If the US NEXRAD products included an SNR field we would use it. In practice, radar constants vary with radar calibration and the estimate of non-hardware noise will also vary depending on conditions and radar siting. So without an SNR field one cannot be quantitatively rigorous.  Rough estimates of SNR (as in**

**the plots above) necessitate assumptions that may well vary among individual radars.**

**At close ranges, RHOHV is reliable for < 20 dBZ.  But, the point of our image muting technique isn't to detect all melting precipitation. The main purpose is to detect melting that is likely to be misinterpreted as heavy snow (> 20 dBZ).**

- L85: A couple comments here, one minor, one major. The minor one is that I think these linear features need to be annotated/circled on the figure so it's clear which linear features you're talking about. If I am correct about the linear features to which you are referring, then here's my major comment: I'm not sure how helpful it is to mute them. The reason why is that this specific line of low CC / high Z (on individual radars, it's often a line that connects a semi-circle of low CC, which is the primary melting layer) represents the edge of the melting layer aloft, where we tend to see the melting layer descend some to the surface because temperatures are only barely above 0 C aloft, extending the melting process and allowing mixed precip to reach closer to the ground. Within this linear band, we very frequently see a zone of sleet pellets at the surface as partially melted snow falls back into a sub-freezing layer and quickly re-freezes into sleet. In fact, your Figure 4 shows this very clearly with both the radar structure and the reanalysis temp x-section. That muted line across southern Long Island is where extremely heavy sleet (i.e., inches of accumulating sleet) was occurring (Fig 6 and related discussion in Picca et al 2014). Do we want to mute such a microphysically important feature that has large implications for surface impacts? If you're just looking for pure snow, I guess it's OK, but I have large concerns about drawing attention away from this feature, at least for an operational meteorologist. Once again, for the public / non-meteorologists, I think this is fine if you wish to present a 'snow map' of sorts, but I have my doubts for forecasters. This is critical information. And if they then have to go look at CC / switch products for clarification on this muted zone, what's the point of the algorithm? I think a sizable explanation is required here to address these concerns with the current design.

**Thank you for your comment about the linear features, we have annotated them on the figure. We used this as an example since many investigators were referring to the linear features as "snowbands" when in fact they are melting precipitation. The primary purpose of this method is to *de-emphasize* regions of mixed precipitation in order to correctly identify regions of heavy snow.  We agree that sleet has important winter weather impacts but its identification is outside of the scope of this paper.**

- L125-132: In Figure 5 for this case, the melting layer is very evident, starting at about 100 km range (except for the collapsed portion to the north). Often at lower elevation angles and more extended ranges, the melting layer presents as a broad area of 'speckly' CC, presumably due to the decreased resolution / increased volume size. With that in mind, pockets of CC > 0.97 often occur in this zone, where perhaps the volume is encountering mainly one precipitation type (e.g., snow just beginning to melt or rain almost entirely finished with the melting process). These zones often are still characterized by higher reflectivity and we see that in Fig 5. Due to the 0.97 threshold, though, much of this melting ring is not muted, showing a shortcoming in this technique. Given this is clearly a zone of higher reflectivity associated with mixed precipitation, the lack of more widespread muting is concerning.

  **Our method is not intended to replace full-featured hydrometeor identification. Rather, our goal is to do one thing, reduce mis-identification of heavy snow, robustly. In this example, the unmuted values that are part of the melting band "arc" are < 20 dBZ so would not be mis-identified as heavy snow.**

Minor Comments:

- L12: Why specify coastal regions? There are plenty of mixed precipitation winter storms across the interior US.

  **In our research we have applied the technique to storms in several geographic locations. The technique can be applied to anywhere mixed precipitation occurs. In the introduction we changed "mid-latitude coastal regions" to "mid-latitude regions" to reflect this.**

  "Winter storms in mid-latitude regions often contain subregions with rain, mixed precipitation, and snow that move and evolve over the storm lifetime (Schultz et al., 2019)." (lines12-13)

- L30-42: Related to my comments above, I wonder if this is truly problematic for trained analysts. Most radar software offers multi-panel views that can show Z and CC (along with other variables) side by side, reducing the need for switching, etc. Additionally, winter weather scenarios evolve on slower timescales, attenuating the cognitive load issue. Would it be better to suggest/emphasize this as a visual tool for presentation to non-meteorologists?

  **We note that viewing products side by side has similar cognitive load challenges as switching back and forth between products (Sweller et al. 2011).**

> **We have added materials to the Conclusions to better describe the intended users who are not operational meteorologists.**

"The proliferation of weather radar web interfaces and mobile apps has made operational radar data easily accessible to a wide range of users with varying levels of radar data interpretation expertise. People who are well versed in the subtle nuances of interpreting weather radar data represent only a subset of research meteorologists and an even smaller subset of the broader set of radar data users which includes emergency managers, TV weathercasters, and airport operators." *(lines 155-158)*

- L53-55: Is it technically more accurate to say "…4 km above radar level" ? Radars located at higher elevations may be scanning above 4 km AGL in some areas within 200 km range, due to land sloping downward from the radar. For instance, I think High Plains radars like FTG are scanning over 4 km AGL within 200 km range to the east.

> **Thank you for catching this. We have changed the text to "above radar level".**

"We include only data within 200 km range from a radar as this is sufficient for combining data from multiple radars in much of the continental US without substantial gaps and constrains the beam center to be below 4 km altitude above radar level." *(lines 64-66)*

- L105: Would clarify that "height of the radar beam (black X in Fig 2b)" refers to height of the 88D data used to construct the regional mapping. You do so in the figure caption but it was confusing as I read the main text because "radar" is ambiguous. At first I thought it referred to the on-board radar, which doesn't make sense of course since that's a nadir-pointing radar.

> **Thank you for the comment. We have updated the text to "…height of the NEXRAD radar beam used to create the regional map…"**

"As the airplane reaches around 175 km in the transect, one can see that the height of the NEXRAD radar beam used to create the regional map (black X in Fig. 2b) begins to intersect the melting layer." *(lines 122-124)*

- L118-124: I suspect some, if not all, of this particular shape in the melting layer structure is because the very close range to radar allows us to resolve this structure much better (and at much lower altitude) than is usual with the 88D network. The end result of this arc-like structure is that we see re-freezing into sleet pellets within that cold air closer to the surface (as I mention above in my major comment). We almost always observe sleet pellets underneath these linear features on the edge of the 0 C isotherm aloft, which would suggest this thermal structure is pretty common and the defined arc structure is more a case of radar resolution, rather than an anomalous thermal environment. See Fig 11d in Griffin et al 2014

https://journals.ametsoc.org/view/journals/wefo/29/6/waf-d-14-00056_1.xml?rskey=9QTt6F&result=1

**Thank you for drawing our attention to this article, we have included a sentence citing the paper and the discussion of the same feature.**

"The structure of the melting layer in this example is also discussed in Griffin et al. (2014)." (lines 144-145)

Figures / Tables

- Table 1: The caption is oddly written. Suggest changing to something like "The correlation coefficient values associated with physical mechanisms that increase snow radar reflectivities when other conditions are held constant." Additionally, I think some further clarification could be necessary. In the first two columns, you are not specifying the nature of the ice particles. If it's a diverse array of crystal types, CC can be lower than 1. While not dramatically lower (let's say 0.95-1), I would say "~1" does not accurately describe such a scenario. Probably should add a condition in which we assume uniform particle habits, if you wish to maintain "~1"

  **Thank you for your comment, we have changed the table caption.**

New table caption reads "Correlation coefficient values associated with physical mechanisms that increase radar reflectivities when Z > 20 dBZ and other conditions are held constant."

  **Also we note Griffin et al. 2000, JAMC, (which we cite in the paper) https://journals.ametsoc.org/view/journals/apme/59/4/jamc-d-19-0128.1.xml?tab_body=fulltext-display paper says that detection of melting layer in weak echo (below 20 dBZ) frequently failed and provides more support for 0.97 rhoHV threshold.**

  **"After testing this method on several events, a threshold of ρhv ≥ 0.97 was found to exhibit the best agreement with the curvature results of FZ95. Overall, the heights of the ML top and ML bottom for the FZ95 curvature and polarimetric methods compare well within regions of higher ZH, with the FZ95 curvature method exhibiting slightly higher (i.e., approximately 200 m) ML tops and slightly higher or lower (i.e., approximately 50 m) ML bottoms. Within regions of ZH < 20 dBZ, the FZ95 method frequently failed."**

- Figure 3: Since the text and the data from Fig 2 only discuss the precipitation types / radar from around 16 UTC, why do you include all of the other times from ASOS data? I think if the text / analysis comprehensively discussed the p-type changes over time at the various sites, it would be more relevant, but as it stands right now, I'm not sure why you present the other times. You do mention the transitions at

KBGM in lines 111-112, but there is no synthesis with the algorithm / radar data. This would be a much better analysis if, for example, you included mosaicked algorithm output around these transition times and compared those data with the ASOS data. Without them, the additional data in Fig 3 are distracting and unnecessary.

**The figure with the ASOS time series, (now Fig. 4), complements the airborne EXRAD radar cross-sections (new Fig. 3) to serve as verification of the position (and timing) of the precipitation transitions.**

"The surface observations and timing of precipitation transitions align well with the evolution and movement of the storm (Fig. 2 and 3)." (lines 127-128)

Other fields from the ER-2 radars to depict the spatial position of the melting layer (new Fig. 3 in manuscript)

[Figure]

References:

Bringi, V. and Chandrasekar, V.: Polarimetric Doppler Weather Radar: Principles and Applications, Cambridge University Press, 2001.

Giangrande, S. E., Krause, J. M., and Ryzhkov, A. V.: Automatic Designation of the Melting Layer with a Polarimetric Prototype of the WSR-88D Radar, Journal of Applied Meteorology and Climatology, 47, 1354–1364, https://doi.org/10.1175/2007JAMC1634.1, publisher:210 American Meteorological Society Section: Journal of Applied Meteorology and Climatology, 2008.

Ivić, I. R.: A Simple Hybrid Technique to Reduce Bias of Copolar Correlation Coefficient Estimates, Journal of Atmospheric and Oceanic Technology, 36, 1813–1833, 2019.

Sweller, J., Ayres, P. L., and Kalyuga, S.: Cognitive Load Theory, New York : Springer, 2011., New York, 2011.

---

## Referee Report (RR1)

**Overall Comments**

I appreciate the authors' efforts in revising the paper, and the added research on SNR by consulting with experts in the field is admirable. The paper has improved from the prior version. However, the authors failed to address my significant concern about the algorithm performance, chiefly that there are weaknesses in this algorithm that diminish its efforts in trying to reduce cognitive load. Due to the thresholding, the algorithm will at times mute gates that are characterized by moderate/heavy snow (see the NBF discussion below), and it will also fail to mute gates that are characterized by high Z from melting snow and thus could be mistaken for heavy snow (see the KDIX case discussion below). Nowhere in the text can I find discussion of these weaknesses, potential solutions, or further development.

I'm not trying to badger the point, and as somebody who has developed polarimetric-based algorithms with a similar goal in mind (reducing load on the analyst while still incorporating the wealth of data from polarimetric variables), I understand very much the challenges. I appreciate your efforts in trying to make this information more usable and accessible for radar analysts in winter weather, and application of image muting seems very promising. I applaud that, and you've opened my eyes to the potential capability there. However, you have to acknowledge the drawbacks and weaknesses in your technique to make this paper publishable in my opinion. This is a major concern for me until you do so.

I'll note that I don't think you need to overhaul the paper or that it would necessarily take that long to address these major concerns. However, I think you need to reasonably address the shortcomings by stating what they are, when they may be more common, and how users should approach these situations.

**Major Comments**

L32-39: I very much appreciate your efforts in digging into radar data quality with range, resultant impacts on SNR, and how that could then feed into your technique. That said, I can state with confidence that reductions in $\rho_{hv}$ due to NBF frequently occur at more distant ranges and is often not masked by the SNR effect with range. And not just with convective / warm-season scenarios, but in winter weather as well. I've even seen it caused by pure heavy snow at S band, presumably from a large concentration of crystals within part of the beam. Granted, pure-snow NBF is pretty rare. However, significant NBF from melting snow, which impacts legitimate heavy-snow gates down-radial, isn't all that uncommon. I've included an example from the 8-9 February 2013 storm over southern New England (0.5deg KOKX 02/09/2013 0025 UTC). Annotated is an NBF corridor of $\rho_{hv} < 0.97$ in many snow gates associated with Z > 20 dBZ (I went through GR myself and sampled 20-30 dBZ in these gates). These gates would be incorrectly muted in your technique.

[Figure]

With this in mind, I don't think it would be all that uncommon for this technique to incorrectly mute moderate/heavy snow gates due to the influence of NBF, especially in more intense winter storms with more pronounced melting layers / NBF. This needs to be addressed, at a minimum acknowledging the weakness and perhaps suggesting possible improvements going forward. Even if at least stating that users will need to be aware of such radial artifacts in the algorithm.

Secondly, and apologies that I wasn't clear enough in my original comments, your response to my comment on what was then L125-32, referring to then Fig 5 (now Fig 6), is incorrect. There are absolutely unmuted pockets of > 20 dBZ within the melting arc for the KDIX case (see the darker, unmuted values over Delaware, for instance, in screengrabs from your figure – added below). It appears they are unmuted as a result of being above the $\rho_{hv}$ threshold, which I understand. But as I mentioned in my original comment, you often can see this occur in the melting layer where either large snow aggregates are just beginning to melt (dielectric constant is up so Z increases but the diversity isn't quite enough to drop $\rho_{hv}$ below 0.97) or you have mainly large drops where the melting process has almost finished.

[Figure]

If the argument is that these gates are so far south in this particular case that a radar analyst would know they can't be heavy snow, then why are we muting other gates nearby (the speckled grays)? Either we should be muting much more of this region or we shouldn't. This is a drawback as currently designed.

Moreover, if we look farther north at the same radar scan time, we find more examples of unmuted high Z gates in the melting arc. I pulled the data from KDIX at 1737 UTC, 01 Dec 2019 and took a quick look in eastern Pennsylvania (attached GR image below). At 0.5 deg overhead the corridor from Allentown to Easton, there are many gates of > 20 dBZ (in fact some 30-35 dBZ) with $\rho_{hv}$ > 0.97, resulting in them being unmuted. For example, overhead the KABE station (red marker in the attached GR image), Z is ~25 dBZ while $\rho_{hv}$ > 0.97. In turn, these are unmuted gates. The KABE observations at this time, understandably, are all freezing rain, as we have melting occurring overhead:

```
ABE,2019-12-01 17:11,KABE 011711Z 08009KT 3SM FZRA OVC011 M01/M04 A2982 RMK
AO2 SFC VIS 4 RAE05FZRAB05 PRESFR P0003 I1001 T10111039
ABE,2019-12-01 17:43,KABE 011743Z 08014KT 3SM FZRA OVC008 M01/M04 A2974 RMK
AO2 SFC VIS 4 RAE05FZRAB05 PRESFR P0006 I1002 T10111039
ABE,2019-12-01 17:51,KABE 011751Z 07015G20KT 3SM -FZRA BKN006 OVC012 M01/M03
A2971 RMK AO2 SFC VIS 5 RAE05FZRAB05 PRESFR SLP066 P0007 60007 I1002 I6002
T10061033 10000 21011 58067
```

From your comments to the reviewers, I now understand that your algorithm is not supposed to be an all-encompassing melting detection. However, this example is classic bright-banding that, based on your stated intentions, should be muted. This deficiency needs to be acknowledged in the manuscript. If we expect and hope for non-expert users to use and trust your technique, then potential pitfalls need to be clearly stated.

[Figure]

Note how many high-Z gates of this melting-layer bright-banding are not muted in your figure (including in the vicinity of KABE, where Z of ~25 dBZ is present with FZRA being reported at the surface). Are users expected to mentally apply speckled muting to other gates? This seems counter-productive to reducing cognitive load. If we don't expect most users to be expert radar analysts, which I agree is a reasonable expectation, then it needs to be pretty clear that all of the inflated Z in this area is from melting; otherwise, I can easily envision non-expert users interpreting these high-Z gates as heavy snow, due to the gates being unmuted. Thus, this must be addressed in the text. I am not saying you need to solve this problem for this manuscript. Rather, you need to acknowledge the problem and suggest possible remedies, improvements, training considerations, and/or avenues for future development to attenuate the issues.

[Figure]

**Minor Comments**:

L28-29: should be 'e.g.' instead of 'i.e.' These are examples of the situations, not other ways of saying them. For instance, mixed precipitation isn't the only situation causing diversity. A mixture of ice crystal habits aloft can reduce $\rho_{hv}$, for example.

Figure 1: Thank you for adding the oval annotations, but the color choice makes them pretty difficult to see (at least for somebody like me with a slight color deficiency). I had to really stare at them. Would suggest a lighter color for the annotations.

L100: Would change to "…could be misinterpreted as *purely* snowbands…" Some parts of these bands are absolutely heavy snow (as indicated by $\rho_{hv}$ and your muting technique).

L113: "through" is misspelled

L119: Since these values would be *more* negative, it should either be < -4 ms-1 or that the magnitude is > 4 ms-1

---

## Author Response (AR2)

**Tomkins et al. (2022) review response #2**

RC2:

**Overall Comments**

I appreciate the authors' efforts in revising the paper, and the added research on SNR by consulting with experts in the field is admirable. The paper has improved from the prior version. However, the authors failed to address my significant concern about the algorithm performance, chiefly that there are weaknesses in this algorithm that diminish its efforts in trying to reduce cognitive load. Due to the thresholding, the algorithm will at times mute gates that are characterized by moderate/heavy snow (see the NBF discussion below), and it will also fail to mute gates that are characterized by high Z from melting snow and thus could be mistaken for heavy snow (see the KDIX case discussion below). Nowhere in the text can I find discussion of these weaknesses, potential solutions, or further development.

I'm not trying to badger the point, and as somebody who has developed polarimetric-based algorithms with a similar goal in mind (reducing load on the analyst while still incorporating the wealth of data from polarimetric variables), I understand very much the challenges. I appreciate your efforts in trying to make this information more usable and accessible for radar analysts in winter weather, and application of image muting seems very promising. I applaud that, and you've opened my eyes to the potential capability there. However, you have to acknowledge the drawbacks and weaknesses in your technique to make this paper publishable in my opinion. This is a major concern for me until you do so.

I'll note that I don't think you need to overhaul the paper or that it would necessarily take that long to address these major concerns. However, I think you need to reasonably address the shortcomings by stating what they are, when they may be more common, and how users should approach these situations.

**Major Comments**

L32-39: I very much appreciate your efforts in digging into radar data quality with range, resultant impacts on SNR, and how that could then feed into your technique. That said, I can state with confidence that reductions in hv due to NBF frequently occur at more distant ranges and is often not masked by the SNR effect with range. And not just with convective / warm- season scenarios, but in winter weather as well. I've even seen it caused by pure heavy snow at S band, presumably from a large concentration of crystals within part of the beam. Granted, pure- snow NBF is pretty rare. However, significant NBF from melting snow, which impacts legitimate heavy-snow gates down-radial, isn't all that uncommon. I've included an example from the 8-9 February 2013 storm over southern New England (0.5deg KOKX 02/09/2013 0025 UTC). Annotated is an NBF corridor of hv < 0.97 in many snow gates associated with Z > 20 dBZ (I went through GR myself and sampled 20-30 dBZ in these gates). These gates would be incorrectly muted in your technique.

[Figure]

With this in mind, I don't think it would be all that uncommon for this technique to incorrectly mute moderate/heavy snow gates due to the influence of NBF,

especially in more intense winter storms with more pronounced melting layers / NBF. This needs to be addressed, at a minimum acknowledging the weakness and perhaps suggesting possible improvements going forward.

Even if at least stating that users will need to be aware of such radial artifacts in the algorithm.

**To make users aware of the potential artifacts we have revised the following text:**

**(lines 32-33) While correlation coefficient is insensitive to radar power calibration, it does suffer from other data quality problems. Artificially lower rhoHV values can occur along radials downrange of sharp gradients in differential phase (Ryzhkov, 2007).**

**(lines 38-44) Unlike radars that transmit at horizontal and vertical polarizations, the NEXRAD radar transmits at a single polarization oriented at 45 degrees which reduces the overall sensitivity of the radar and in conditions with canted, oriented ice can reduce the correlation coefficient (Rauber and Nesbitt, 2018). In practice, the impact of SNR tends to be much more prevalent than non-uniform beam filling. This suggests that the SNR effect masks most of the effects of non-uniform beam filling in NEXRAD correlation coefficient data quality. Dual polarization radar variable data quality problems are more pronounced when there are mismatched antenna patterns in the horizontal and vertical polarizations (Bringi and Chandrasekar, 2001) which are more common in operational radars than research radars.**

**(lines 95-100) Any method relying on a particular variable as input will not work well when there are data quality problems with that variable. Data quality problems with correlation coefficient along radials downrange of sharp gradients in differential phase will yield sporadic image muted areas radial to the radar that will not move consistently with the advection of locally enhanced reflectivity bands within the storm. Regions of speckled image muting based on the method described here could either be a result of small spatial scale variations in the melting of snow or noise in the**

**correlation coefficient field related to low signal to noise ratios which are more common at farther ranges from the radar (Ivić, 2019).**

**(lines 193-195) Enhanced reflectivity bands that are snow or contain mixed precipitation will generally move consistently with the advection of other reflectivity features rather than being fixed either concentrically or radially to the radar position.  Hence, our image muting method is best used as part of movie loop sequences rather than as individual images.**

Secondly, and apologies that I wasn't clear enough in my original comments, your response to my comment on what was then L125-32, referring to then Fig 5 (now Fig 6), is incorrect. There are absolutely unmuted pockets of > 20 dBZ within the melting arc for the KDIX case (see the darker, unmuted values over Delaware, for instance, in screengrabs from your figure – added below). It appears they are unmuted as a result of being above the rhv threshold, which I understand. But as I mentioned in my original comment, you often can see this occur in the melting layer where either large snow aggregates are just beginning to melt (dielectric constant is up so Z increases but the diversity isn't quite enough to drop rhv below 0.97) or you have mainly large drops where the melting process has almost finished.

[Figure]

If the argument is that these gates are so far south in this particular case that a radar analyst would know they can't be heavy snow, then why are we muting

other gates nearby (the speckled grays)? Either we should be muting much more of this region or we shouldn't. This is a drawback as currently designed.

Moreover, if we look farther north at the same radar scan time, we find more examples of unmuted high Z gates in the melting arc. I pulled the data from KDIX at 1737 UTC, 01 Dec 2019 and took a quick look in eastern Pennsylvania (attached GR image below). At 0.5 deg overhead the corridor from Allentown to Easton, there are many gates of > 20 dBZ (in fact some 30-35 dBZ) with rhv > 0.97, resulting in them being unmuted. For example, overhead the KABE station (red marker in the attached GR image), Z is ~25 dBZ while rhv > 0.97. In turn, these are unmuted gates. The KABE observations at this time, understandably, are all freezing rain, as we have melting occurring overhead:

```
ABE,2019-12-01 17:11,KABE 011711Z 08009KT 3SM FZRA OVC011
M01/M04 A2982 RMK AO2 SFC VIS 4 RAE05FZRAB05 PRESFR P0003 I1001
T10111039
ABE,2019-12-01 17:43,KABE 011743Z 08014KT 3SM FZRA OVC008
M01/M04 A2974 RMK AO2 SFC VIS 4 RAE05FZRAB05 PRESFR P0006 I1002
T10111039
ABE,2019-12-01 17:51,KABE 011751Z 07015G20KT 3SM -FZRA BKN006
OVC012 M01/M03 A2971 RMK AO2 SFC VIS 5 RAE05FZRAB05 PRESFR
SLP066 P0007 60007 I1002 I6002 T10061033 10000 21011 58067
```

From your comments to the reviewers, I now understand that your algorithm is not supposed to be an all-encompassing melting detection. However, this example is classic bright-banding that, based on your stated intentions, should be muted. This deficiency needs to be acknowledged in the manuscript. If we expect and hope for non-expert users to use and trust your technique, then potential pitfalls need to be clearly stated.

[Figure]

Note how many high-Z gates of this melting-layer bright-banding are not muted in your figure (including in the vicinity of KABE, where Z of ~25 dBZ is present with FZRA being reported at the surface). Are users expected to mentally apply speckled muting to other gates? This seems counter-productive to reducing cognitive load. If we don't expect most users to be expert radar analysts, which I agree is a reasonable expectation, then it needs to be pretty clear that all of the inflated Z in this area is from melting; otherwise, I can easily envision non-expert users interpreting these high-Z gates as heavy snow, due to the gates being unmuted. Thus, this must be addressed in the text. I am not saying you need to solve this problem for this manuscript.

Rather, you need to acknowledge the problem and suggest possible remedies, improvements, training considerations, and/or avenues for future development to attenuate the issues.

[Figure]

**We have added the following text to the manuscript:**

**(lines 165-169) Users should use caution interpreting features at longer ranges from the radar where RHOHV suffers from quality issues related to low signal to noise ratio. For example, in Figure 6, the speckled muting beyond 100 km range of the radar is likely the result of the superposition of an increase in correlation coefficient associated with low signal to noise ratio and a decrease associated with melting. The animation of this figure in the Supplement illustrates that the concentric speckled region remains approximately stationary to the radar and hence can be visually distinguished from advecting reflectivity bands.**

**(lines 200-205) The method to detect melting regions is not perfect in large part since such algorithms are limited by the input data quality. For U.S. NEXRAD data, without improvements in the data quality of RHOHV, detection of melting regions particularly at farther ranges will be more speckled than at closer ranges. If the signal to noise ratio field is made available it can be used to filter out questionable RHOHV values and improve the detection of melting regions. Users are advised to utilize movie loops to assess the time and spatial continuity when distinguishing**

**band-like enhanced reflectivity features corresponding to heavy snow bands from those that include melting.**

**Minor Comments:**

L28-29: should be 'e.g.' instead of 'i.e.' These are examples of the situations, not other ways of saying them. For instance, mixed precipitation isn't the only situation causing diversity. A mixture of ice crystal habits aloft can reduce rhv, for example.

**Thank you for catching this. We have updated the text in the manuscript.**

**(lines 27-30) Correlation coefficient is approximately one in regions with single hydrometeor types (e.g. only rain or only snow) and decreases in regions where there is an increasing diversity of hydrometeor orientations and shapes (e.g. mixed precipitation such as rain with snow and/or partially melted ice) (Giangrande et al. 2008, Rauber and Nesbitt, 2018).**

Figure 1: Thank you for adding the oval annotations, but the color choice makes them pretty difficult to see (at least for somebody like me with a slight color deficiency). I had to really stare at them. Would suggest a lighter color for the annotations.

**Thank you for your comment. We have updated the annotations to be white.**

L100: Would change to "...could be misinterpreted as *purely* snowbands..." Some parts of these bands are absolutely heavy snow (as indicated by rhv and your muting technique).

**We have added the suggestion to the text.**

**(lines 110-111) This example shows two linear features in central New York that could be misinterpreted as purely snowbands when analyzing the reflectivity alone (white ovals in 1a)**

L113: "through" is misspelled

**Thank you for catching this. We have updated the text in the manuscript.**

**(line 123-126) The gray region in the image muted regional map indicates a quasi-linear region of mixed precipitation extending through eastern New York up to Vermont and New Hampshire (Fig. 2a) between areas of primarily snow (to the northwest in upstate New York) and primarily rain (to the southeast over southern New England).**

L119: Since these values would be *more* negative, it should either be < -4 ms-1 or that the magnitude is > 4 ms-1

**Thank you for catching this. We have updated the text in the manuscript.**

**(line 130) Under the melting layer, the values of downward pointing Doppler velocity < -4 ms$^{-1}$ indicate the rain layer.**